# Pre-COVID-19 knowledge, attitude and practice among nurses towards infection prevention and control in Bangladesh: A hospital-based cross-sectional survey

**Md. Golam Dostogir Harun** [1,2]*, **Md Mahabub Ul Anwar**[3], **Shariful Amin Sumon**[2], **Md Abdullah-Al-Kafi**[4], **Kusum Datta**[5], **Md. Imdadul Haque**[2], **A. B. M. Alauddin Chowdhury**[2], **Sabrina Sharmin**[6], **Md Saiful Islam**[7]

**1** Infection Disease Division, icddr,b, Dhaka, Bangladesh, **2** Department of Public Health, Daffodil International University, Dhaka, Bangladesh, **3** Department of Population Sciences, University of Dhaka, Dhaka, Bangladesh, **4** School of Population and Public Health, University of British Columbia, Vancouver, British Columbia, Canada, **5** Dhaka Medical College Hospital, Dhaka, Bangladesh, **6** Department of Public Health, Bangabandhu Sheikh Mujib Medical University (BSMMU), Dhaka, Bangladesh, **7** University of New South Wales, Sydney, Australia

* dostogirharun@gmail.com

**Data Availability Statement:** Data cannot be shared publicly because of the data sharing policy

## Abstract

### Introduction

Hospital-acquired infections endanger millions of lives around the world, and nurses play a vital role in the prevention of these infections. Knowledge of infection prevention and control (IPC) best practices among nurses is a prerequisite to maintaining standard precautions for the safety of patients.

### Aim

The study aims to assess knowledge, attitudes, and practices (KAP) towards IPC including associated factors among the nurses of a tertiary care hospital in Bangladesh.

### Methods

We conducted this hospital-based cross-sectional study from October 2017 to June 2018 at Dhaka Medical College Hospital among 300 nurses working in all departments. We calculated three KAP scores for each participant reflecting their current state of knowledge and compliance towards IPC measures. Descriptive, bivariate and multivariable analyses were conducted to determine KAP scores among nurses and their associated factors.

### Results

Average scores for knowledge, attitudes, and practices were 18.6, 5.4, and 15.5 (out of 26, 7, and 24), respectively. The study revealed that the majority (85.2%) of the nurses had a good to moderate level of knowledge, half (51%) of them showed positive attitudes, and only one fifth (17.1%) of the nurses displayed good practices in IPC. The respondents'

for the study hospital. Data are available from the corresponding author/ Institutional Data Access and Ethics Committee. Please contact Md. Golam Dostogir Harun (dostogirharun@gmail.com) for researchers who meet the criteria for access to confidential data. Data will also be available at the department of Public Health, Daffodil International University, Dhaka Bangladesh (headph@daffodilvarsity.edu.bd) upon request.

**Funding:** The authors received no specific funding for this work.

**Competing interests:** The authors have declared that no competing interests exist.

age, education, monthly income and years of experience were found to have statistical associations with having moderate to adequate level of KAP scores. Aged and experienced nurses were found more likely to have poor knowledge and unfavorable attitude toward IPC practices.

## Conclusion

The majority of nurses had good IPC knowledge, but their practices did not reflect that knowledge. In particular, nurses needed to improve the proper IPC practice for better patient care and to protect themselves. Regular IPC training and practice monitoring can enhance the IPC practice among nurses.

## Introduction

Hospital-acquired infections (HAI) are considered a major global health problem that endangers millions of lives every year across the globe [1]. Annually, two million people worldwide are affected by different HAIs, out of whom approximately 100,000 die from such infections [2]. Nowadays, HAIs are increasing alarmingly in low- and middle-income countries (LMICs), with up to 25% of hospitalized patients affected, which is 2–20 times higher than in developed countries [3, 4]. Around 75% of the global HAI burden is borne by LMICs [5]. Studies found that HAIs cause longer hospital stays, substantial morbidity and mortality, antimicrobial resistance (AMR), and high economic and productivity loss [4, 6]. It has been estimated that HAIs could result in a loss of about US$100 trillion by 2050, as well as an additional 10 million deaths related to AMR in hospital settings [7, 8]. Healthcare-associated infections also impact the health and well-being of healthcare professionals. Around 40% of Hepatitis B and C infections in healthcare providers occur due to occupational exposures exposure that could be avoided through adherence to IPC measures [9, 10]. Nurses are the frontline health workers for hospital infection control, as well as in the protection of patients, visitors, and other staff [11]. Beyond the spectrum of designated duties, nurses render various additional services to help sustain healthcare delivery [12]. They are also the most vulnerable to infection with various HAIs and transmitting them to patients [13, 14]. Studies have found that adequate knowledge and practice of different components of IPC among nurses can reduce the burden of HAIs by 30–70% with the help from some cost-effective and feasible strategies [4, 15, 16].

For LMICs like Bangladesh, HAIs are a massive problem in healthcare settings, and major deficits in infection control lie in the availability of essential resources, adequately trained personnel, and IPC compliance [17]. Also, there is a scarcity and inconsistency of available data on rates of HAIs and risk factors [18]. The majority of common HAIs are transmitted by healthcare workers in Bangladesh, especially nurses, due to a lack of knowledge related to infection control and failure to consistently and properly practice IPC measures [9, 19]. Moreover, Bangladesh is considered a regional hotspot for emerging infectious disease threats, such as severe acute respiratory syndrome (SARS) and Nipah virus infection [20]. The evidence regarding IPC practice among healthcare workers in Bangladesh is limited, so it is crucial to identify gaps and generate evidence on IPC compliance for developing effective interventions to control the rate of HAIs [21]. This study aimed to assess the pre-COVID-19 infection control practices among nurses in the largest tertiary care hospital in Bangladesh. Findings from

this study identify system loopholes and opportunities for improving universal precautions against HAIs in both government and private healthcare settings.

## Methods

### Study design and setting

This hospital-based cross-sectional study was carried out at Dhaka Medical College Hospital (DMCH) from October 2017 to June 2018. This is the largest tertiary hospital in Bangladesh with a 2,600 bed capacity, an annual inpatient turnover of approximately 176,500, and the modern diagnostic and investigation technologies to provide both inpatient and outpatient treatment and care support [22]. No active infection prevention and control program was running during the study period to assess the current infection level [23] and take mitigation strategies.

### Study participants and sampling procedure

The study population includes all nurses working at DMCH during the study period. Previous study reviews found that the prevalence of IPC-related good knowledge level was 71% and positive attitude was 66% among nurses in Bangladesh [24, 25]. Considering this prevalence of knowledge and positive attitude along with 95% confidence interval (CI), 5.5% absolute precision and 5% non-response rate, we calculated the sample size separately for knowledge, attitudes, and practices and selected the highest sample size, which is 300 for this study. We prepared a list of 1,865 nurses alphabetically based on information provided by DMCH human resource administration and used it as a sampling frame (Fig 1).

Then we used a systematic sampling procedure (every sixth nurse [k = N/n = 1865/311]) and selected 311 nurses to become study participants. Among them, after considering refusal, we interviewed 300 participants by explaining the purpose of the study. The detailed sampling procedure is shown in Fig 1. We obtained written informed consent from each of the participants and interviewed them for 30–45 minutes using a structured questionnaire. Participants were assigned an anonymous ID to maintain confidentiality.

### KAP questions on IPC

A structured questionnaire was developed focusing on the research objectives and target population, based on literature reviews and expert opinions [23, 26]. After reviewing the literature [27] and evaluating the questionnaire, 57 questions were finalized with 26 knowledge-, 7 attitude-, and 24 practice-related questions. This questionnaire was pretested among the 20 non-sampling nurses (who were excluded from the sampling frame) using the Bangla version of the questionnaire to get feedback on the suitability, appropriateness, and sequencing of the questions. We addressed the feedback, updated the questionnaire, and conducted the KAP survey among nurses. Knowledge-related questions primarily focused on the concepts of HAI transmission, IPC activities, disinfection, and sterilization. Each question was scored 1 for a correct response and 0 for an incorrect answer. The total knowledge score was 26. Questions related to attitude mainly focused on the nurses' approach towards infection control policies and procedures, washing hands, and wearing personal protective equipment (PPE). Responses demonstrating a positive attitude received a score of 1, and negative responses a score of 0. The attitude dimension contained a total score of 7. Similarly, questions for measuring practices were related to compliance with universal precautionary guidelines, attending IPC training, maintaining IPC guidelines. Each question was scored 1 for correct practice, and the total score for the practice dimension was 24.

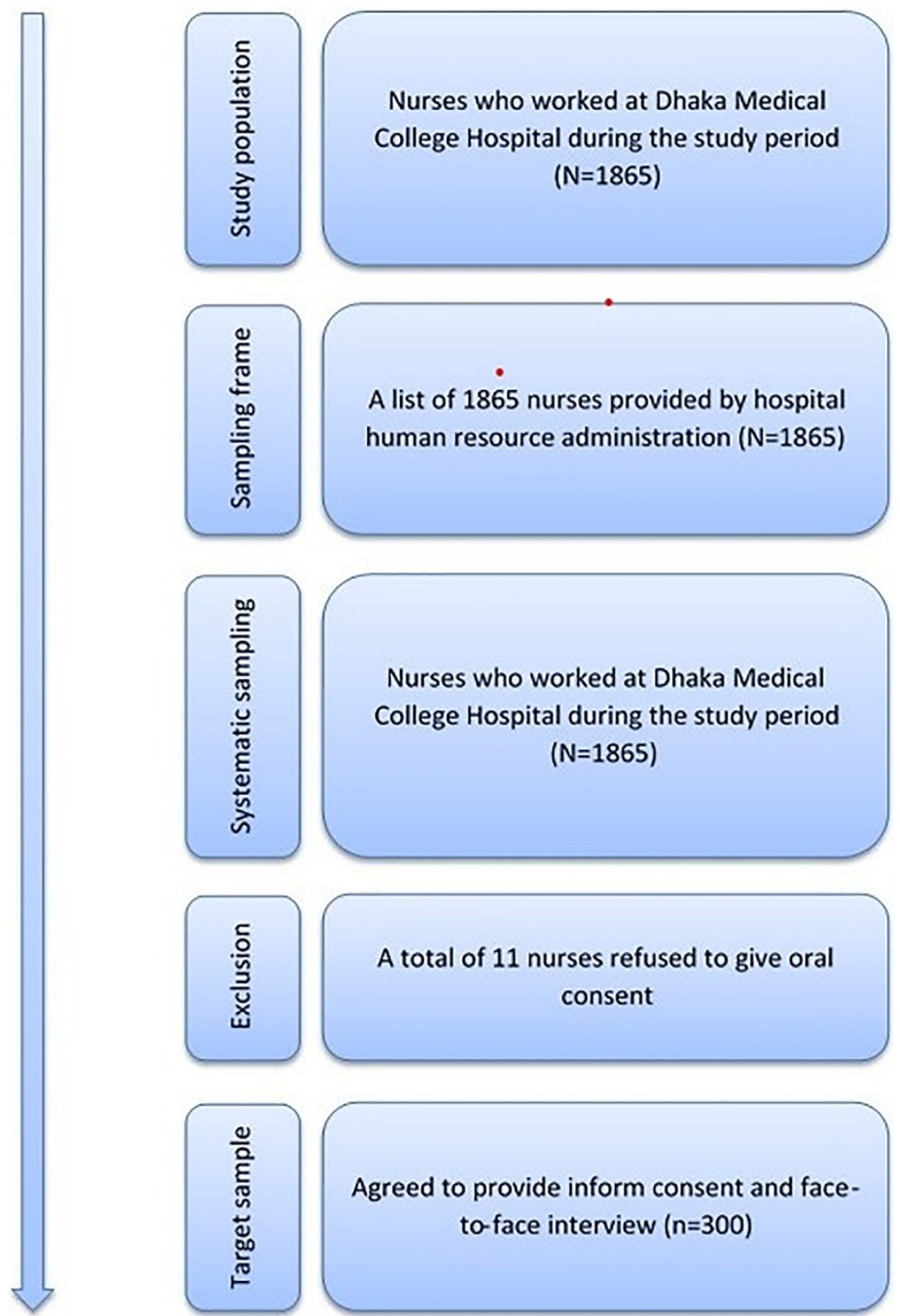

**Fig 1. Sample size and sampling strategy.**

Based on existing literature, and scored the KAP, a participant who scored <60% of the total score was labelled as having a poor KAP level. Similarly, obtaining 60 to 79% of the total score was marked as having a fair KAP level and having a score >80% was considered good [16].

Finally, the participants attaining fair or good scores for each component of the KAP (60–100%) were considered to have average knowledge, a favourable attitude, and safe practice and

were coded as 1 and those whose overall score was < 60%, were coded as 0. Cronbach alpha coefficient values for knowledge, attitudes, and practices related questionnaires content were 0.733, 0.701 and 0.735, respectively. In addition, the Spearman-Brown split-half reliability coefficient values for knowledge, attitude and practice related questionnaire content were 0.857, 0.835 and 0.728, correspondingly.

## Statistical analysis

We summarized the responses to the KAP questionnaire using frequency, percentage, and mean with standard deviation (SD). Additionally, depending on the distribution of the KAP, the Kruskal-Wallis equality of population rank test and Mann-Whitney test were utilized to examine the different scores by socio-demographic variables. To explore potential covariates, we hypothesized that different socioeconomic and demographic variables influence nurses' IPC-related behaviour through enhancing their knowledge, attitudes, and practices. We conducted separate bivariate logistic regression to explore the factors associated with average knowledge, favourable attitude, and safe practice among nurses toward infection prevention and control. Finally, we applied multivariable logistic regression to get the adjusted effect of potential factors on the outcome variables. We reported both unadjusted odds ratios (UOR) and adjusted odds ratios (AOR) with 95% confidence intervals. All the statistical analyses were conducted using Stata 15 software (Stata Corp. 2017).

## Ethical approval and consent to participate

Ethical approval to conduct the study was taken from Daffodil International University. Respective authorities of the study site were contacted before the beginning of data collection and necessary approvals were taken. Written informed consents were taken from the respondents before the study. The goals and objectives of the study were coherently mentioned in the consent paper. The respondents were kept assured about the anonymity of their identity and also about the privacy and confidentiality of the data they would provide. They also had been informed about the voluntary pattern of participation and they were at liberty to leave the survey at any time if they found it not convenient to carry forward. No monetary incentives were provided to the respondents.

## Results

### General characteristics of the nurses

The mean age of the study participants was 34.0 (SD = 9.72) years, and most (83.7%) were female. Approximately three-fourths (75.3%) of participants were married. A majority of nurses (66.7%) had a three years diploma in nursing, 24.3% had a Bachelor of Science in nursing, and only 9% had a Master/MPH or higher degree. About half of the nurses (48%) had $\leq 2$ years of working experience. The average monthly family income was 50,500 BDT (US$600) and the individual average monthly income was BDT 28,888 (US$344) (Table 1).

### Categorization of KAP Score among the nurses

About one-third (30.2%) of nurses demonstrated good knowledge of IPC, but more than half (55%) had only a fair level of knowledge. Regarding the attitudes dimension, more than half (51%) of the nurses displayed a good attitude towards following different aspects of IPC, but one-third (32%) had a poor attitude towards IPC. In regard to practices, only 17.1% of the nurses were found to maintain good IPC practices on hospital premises, with about half (44.1%) of the nurses reporting poor practices (Fig 2).

**Table 1. Characteristics of nurses in Dhaka Medical College Hospital, Bangladesh (N = 300).**

| Background characteristics | n (%) |
|---|---|
| **Total** | **300 (100)** |
| **Age in years** | |
| Mean (SD) | 9.7 (34.0) |
| ≤26 | 83 (27.7) |
| 27–30 | 85 (28.3) |
| 31–42 | 61 (20.3) |
| ≥43 | 71 (23.7) |
| **Sex** | |
| Female | 251 (83.7) |
| **Marital status** | |
| Married | 226 (75.3) |
| Unmarried | 63 (21.0) |
| Other (diverse or widow) | 11 (6.7) |
| **Religion** | |
| Muslim | 222 (74.0) |
| Hindu | 69 (23.0) |
| Christian | 9 (3.0) |
| **Education** | |
| Diploma in nursing | 200 (66.7) |
| BSc in nursing | 73 (24.3) |
| Masters and above | 27 (9.0) |
| **Monthly income (Taka)** | |
| Median (IQR) | 28,888 (28,000–40,000) |
| ≤22400 | 40 (13.3) |
| 22401–33000 | 163 (54.3) |
| 33001–44000 | 33 (11.0) |
| 44001–55000 | 44 (14.7) |
| ≥55001 | 20 (6.7) |
| **Monthly family income (Taka)** | |
| Median (IQR) | 50,500(41,500–100,000) |
| ≤22400 | 27 (9.0) |
| 22401–33000 | 25 (8.3) |
| 33001–44000 | 27 (9.0) |
| 44001–55000 | 75 (25.0) |
| ≥55001 | 146 (48.7) |
| **Total years of working experience** | |
| ≤2 | 144 (48.0) |
| 3–10 | 47 (15.7) |
| 11–18 | 33 (11.0) |
| 19–26 | 44 (14.7) |
| ≥27 | 32 (10.7) |

## Knowledge, attitudes, and practices of nurses towards IPC

**Knowledge of nurses regarding IPC.** The average knowledge score of the nurses was 18.6 of 26 with a SD of 2.8. Among all nurses, 14.8% had poor knowledge (score 14–15) and the remaining 85.2% had an average level of knowledge (score ≥ 16) (compiling fair and good categories) (Fig 3).

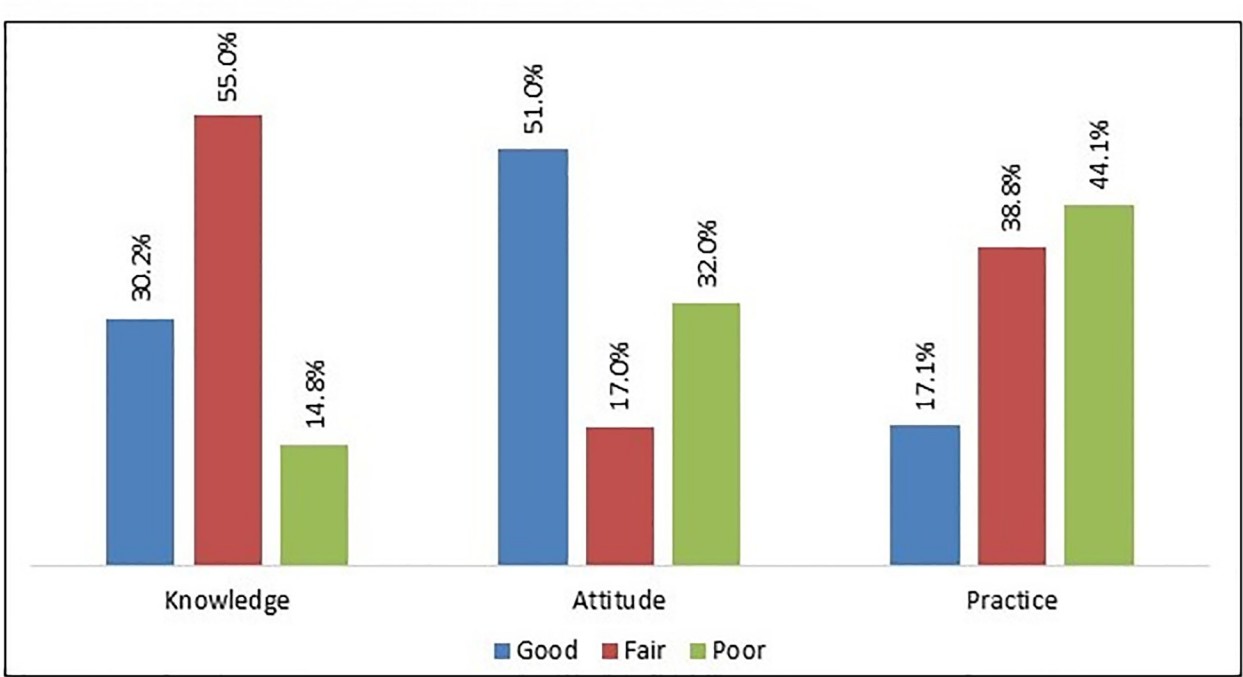

**Fig 2. Percentage of different categories of KAP of nurses towards IPC.**

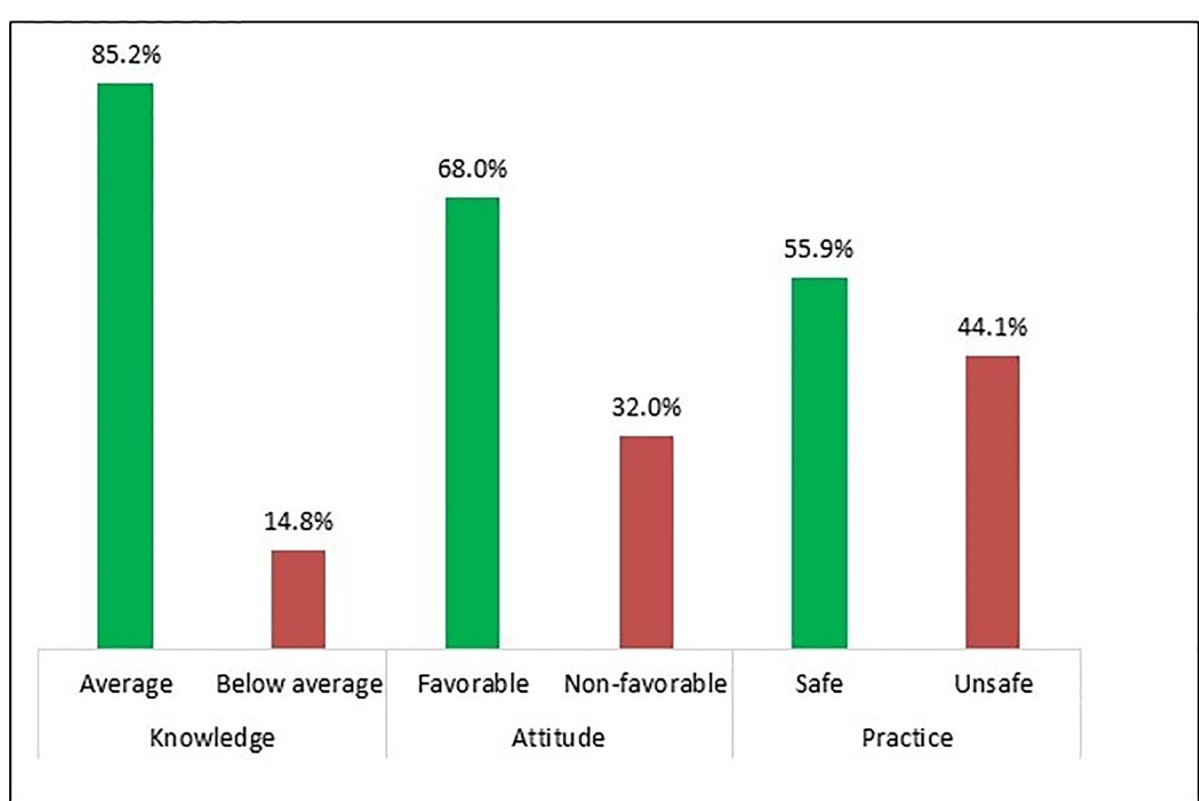

**Fig 3. Percentage of distribution of two main categories of KAP of nurses towards IPC at DMCH in Bangladesh.**

Approximately half (53%) of the participants correctly answered about the transmission of HAIs. Almost all the nurses correctly answered about familiarity with and importance of IPC guidelines (98.3%), adherence to IPC procedures (98.7%) and application of sterilization to reduce the risk of infection (99.3%). On the contrary, a low percentage of correct answers were found for knowledge questions related to the resources needed to comply with infection prevention guidelines (23.3%), bending used needles (4.3%), drenching in glutaraldehyde for 10 hours at 20–35˚C for decontamination (29.0%) and compliance with IPC guideline even in heavy workload (40.3%) (Table 2).

**Attitudes of nurses towards IPC.**   Nurses' average attitudes score was 5.4 of 7 (SD = 1.5). Among them, about two-thirds (68.0%) showed favourable attitudes (score ≥5) towards IPC (Fig 3).

**Table 2. Knowledge of the nurses regarding infection prevention and control, universal precautions, disinfection, sterilization (N = 300).**

| Questions related to knowledge | Correct | Incorrect |
|---|---|---|
|  | %(n) | %(n) |
| *Hospital-acquired infections (HAI)* | | |
| Hospital-acquired infections (HAI) can be transmitted by medical equipment such as syringes, needles, catheters, thermometers, stethoscopes, etc. | 53.0(159) | 47.0(141) |
| Nosocomial infection is an infection that the patient comes with from home | 56.0(168) | 44.0(132) |
| If there are limited beds available, patients with an infectious disease may be admitted to the same ward with other patients | 80.7(242) | 19.3(58) |
| *IPC guidelines and training* | | |
| I am familiar with hospital-acquired infection guidelines | 97.7(293) | 2.3(7) |
| Policies and procedures for infection control should be adhered to at all times | 98.7(296) | 1.3(4) |
| Infection prevention guidelines are essential to this hospital | 98.3(295) | 1.7(5) |
| My responsibility is not to comply with hospital-acquired infection guidelines | 55.0(165) | 45.0(135) |
| I am familiar with resources/ supplies (ABHS, soap, PPE etc.) needed to comply with infection prevention guidelines | 23.3(70) | 76.7(230) |
| All staff working in the hospital should follow the IPC instructions and perform IPC practice | 50.7(152) | 49.3(148) |
| Standard infection control policies and guidelines are enough to control HAIs | 63.3(190) | 36.7(110) |
| Despite the heavy workload, I should comply with infection prevention guidelines | 40.3(121) | 59.7(179) |
| I should attend in-service training/ related to infection control regularly | 70.3(211) | 29.7(89) |
| *Universal precaution* | | |
| I know the World Health Organization's five moments of hand hygiene. | 63.0(189) | 37.0(111) |
| Only clean water is enough to destroy micro-organisms. | 69.7(209) | 30.0(91) |
| Bathing every day after hospital duty is a universal precaution | 90.0(270) | 10.0(30) |
| Standard precautions apply to all patients regardless of their diagnosis | 82.7(248) | 17.3(52) |
| I know how to prevent and control hospital-acquired infections. | 98.7(296) | 1.3(4) |
| Workplace risk assessment is essential for occupational safety. | 54.0(162) | 46.0(138) |
| I am aware that patients expect me to wash my hands before and after touching them. | 92.7(278) | 7.3(22) |
| Hands should be washed with soap/sanitized after using gloves. | 98.3(295) | 1.7(5) |
| A contaminated item soaked in glutaraldehyde for 10 hours at 20–35˚C is sterilized | 29.0(87) | 71.0(231) |
| A non-correct application of the disinfection/sterilization procedures increases the risk of infection in personnel | 99.3(298) | 0.7(2) |
| The stethoscope must be cleaned/sanitized with alcohol swab pad before and after every patient examine | 94.3(283) | 5.7(17) |
| Items used during a surgical practice should always be sterilized | 98.7(296) | 1.3(4) |
| A non-correct application of the disinfection/sterilization procedures increases the risk of infection in patients | 98.7(296) | 1.3(4) |
| Used needles should never be bent or recapped | 4.3 (13) | 95.7(287) |

**Table 3. Attitude of the nurses toward infection prevention and control, universal precautions, disinfection, sterilization (N = 300).**

| Questions related to attitudes | Positive | Negative |
|---|---|---|
| | %(n) | %(n) |
| Policies and procedures for infection control should be adhered to at all times. | 99.7(299) | 0.3(1) |
| The hospital should have standard infection control policies and guidelines. | 46.3 (139) | 53.7(161) |
| I feel that needles should always be recapped after use and before disposal. | 40.7(122) | 59.3(178) |
| Washing hands with soap or sanitizer before and after patient contact can limit infection transmission. | 93.0(279) | 7.0(21) |
| Warning gloves and a mask every time while taking blood/swabs or handling potentially infectious material. | 75.3(226) | 24.7(74) |
| Using puncture-proof containers for disposing of medical waste. | 65.7(197) | 34.3(103) |
| Aprons/gowns should always be worn to avoid direct contact with blood or body fluids. | 99.7(299) | 0.3(1) |

All the nurses demonstrated a positive attitude towards adherence to IPC policies and procedures and should always wear aprons or gowns (99.7%). The lowest favourable attitude was found for adhering to proper needle disposal procedures (46.3%), as the recapping of needles was commonly considered an acceptable practice (Table 3).

**Practices of nurses towards IPC.** The mean practice score was 15.5 of 24 (SD = 4.0). Less than half (44.1%) of the participants had unsafe/poor practice (score ≤ 14) and the rest of them were classified in the safe level of IPC practice (compiling fair and good categories) (Fig 3). Almost all the nurses provided a "yes" answer to the practice questions related to handwashing after the removal of gloves (96.0%) and disposal of the full sharps box (98.0%). The opposite scenario was found for questions related to proper disposal of needles after giving an injection (28.3%), always wearing a disposable facemask in the possibility of splash or splatter (36.0%) and attending in-service IPC workshop (13.7%) (Table 4).

## Relationship between nurses' characteristics and KAP scores

Female nurses showed a more favourable attitude and safer practices towards IPC when compared to male nurses (p-values < 0.05). Knowledge and attitude towards IPC differed significantly by level of education (p-values < 0.05). Nurses having diploma degrees were found to be more knowledgeable and showed a more positive attitude than the nurses having bachelor's and Master's degrees (p-value < 0.05). A similar significant association of KAP was found with nurses' monthly income (p-value < 0.05). Also, the IPC attitude score was found decrease with increasing years of experience, when comparing those with <2 years of experience to those with 2–18 years (p-value < 0.05) (Table 5).

## Factors associated with knowledge, attitude and practice

The potential factors associated with nurses' average knowledge, a favourable attitude and safe practices towards IPC are shown in Table 6. We found that older nurses were more reluctant towards IPC practices, as nurses aged 31–42 years and those ≥43 years were 60% (AOR = 0.4, 95% CI: 0.2–1.0) and 90% (AOR = 0.1, 95% CI: 0.0–0.2) less likely to practice IPC compared to the reference category (age ≤26 years). The odds of having average knowledge towards IPC were progressively decreasing among nurses with B.Sc (AOR = 0.3, 95% CI: 0.1–0.5) and Masters/above (AOR = 0.2, 95% CI: 0.1–0.7) compared to nurses with only a diploma degree. Nurses with a long time of working experience (27 years plus) had unfavourable attitudes towards IPC practices. Nurses whose monthly family income were ranged from (33001–

**Table 4. Practice of the nurses regarding infection prevention and control, universal precautions, disinfection, sterilization (N = 300).**

| Questions related to practices | Yes | No |
|---|---|---|
| | %(n) | %(n) |
| **IPC guidelines and training** | | |
| Knowledge of infection control is being monitored in the hospital. | 61.0 (183) | 39.0(117) |
| I maintain Policies and procedures for infection control strictly at all times. | 76.3(229) | 23.7(71) |
| I attend in-service training/workshops related to infection control regularly. | 80.0(240) | 20.0(60) |
| I maintain infection control policies and guidelines are enough in the hospital. | 42.7(128) | 57.3(172) |
| As an HCW, I am vaccinated against common pathogens. | 50.3(151) | 49.7(149) |
| My infection prevention & control practices are monitored regularly in the hospital | 55.7(167) | 44.3(133) |
| I attended in-service training/workshop on infection prevention & control last year. | 13.7(41) | 86.3(259) |
| **Universal precautions** | | |
| I always wash my hands with soap or sanitize them before and after direct contact with the patient. | 57.7(173) | 42.3(127) |
| I always put on a mask & glasses when performing invasive and body fluid procedures | 49.3(148) | 50.7(152) |
| I bathe every day after spending time/ performing duty in suspected areas in the hospital to avoid infection. | 73.7(221) | 26.3(79) |
| I always kept used needles or disposable scalpels and blades in the sharps box. | 67.0(201) | 33.0(99) |
| I wash my hands with soap and water after caring for patients. | 61.7(185) | 38.3(115) |
| I wear gloves when touching blood, body fluids, mucous membranes, or non-intact skin. | 82.7(248) | 17.3(52) |
| I wore gloves when I was exposed to deep body fluids or blood products. | 79.7(239) | 20.3(61) |
| I cover my wound with a waterproof dressing before caring for patients. | 83.7(251) | 16.3(49) |
| I wash my hands with soap or sanitize them immediately after removing my gloves. | 96.0(288) | 4.0(12) |
| I kept heavily bloodstained materials in a red plastic bag, irrespective of infectious patient status. | 45.7(137) | 54.3(163) |
| Staff clean up blood spills immediately using the disinfectant | 88.3(265) | 11.7(35) |
| I decontaminate surfaces and devices after use, as a splash or splatter is possible. | 90.0(270) | 10.0(30) |
| I always wear a disposable mask whenever there is a possibility of a splash or splatter. | 36.0(108) | 64.0(192) |
| I wear a gown/apron if soiling with blood or body fluids is likely to be fallen. | 63.3(190) | 36.7(110) |
| I dispose of needles properly after pushing an injection. | 28.3(85) | 71.7(215) |
| The sharps box should be disposed of only when it is full. | 98.0(294) | 2.0(6) |
| I wear eye shield/goggles when I may be exposed to the splashing of bloody discharge/fluid. | 42.3(127) | 57.7(173) |

44000) Taka were more likely to show favourable attitude towards IPC (≤22400 vs. 33001–44000, AOR = 5.3, 95% CI:1.4–20.4). However, nurses with higher monthly income showed to perform more safe practices toward IPC (Table 6).

## Discussion

HAI is an emerging concern in spreading infection, especially due to the escalating trend of infectious diseases. In hospital settings, nurses are the closest healthcare workers to the patients. Having adequate knowledge of HAI burdens will enhance the practice of good IPC behaviour in their everyday tasks. We conducted this study in the largest tertiary care hospital in Bangladesh to investigate the current situation of KAP towards IPC among the nurses. We explored the prevalence of KAP among the nurses as well as identified the potential associated factors influencing compliance towards IPC practices.

The overall mean knowledge score of the nurses was 18.6 out of 26. About 97% of the nurses were familiar with IPC guidelines, policies and procedures, but most of them had inadequate knowledge about the IPC resource availability to comply with IPC guidelines. Other similar

**Table 5. Relationship of average knowledge, favourable attitudes and safe practices scores with participant characteristics, 2017–2018, Dhaka Medical College Hospital, Bangladesh (N = 300).**

| Characteristics | Average knowledge | | Favorable attitude | | Safe practice | |
|---|---|---|---|---|---|---|
| | % (n) | p-value | % (n) | p-value | % (n) | p-value |
| **Overall** | 85.3(256) | | 68.0(204) | | 55.9 (147) | |
| **Age in years** | | | | | | |
| ≤26 | 26.2(67) | | 26.0(53) | | 30.6(45) | |
| 27–30 | 30.5(78) | 0.188 | 30.9(63) | 0.359 | 32.0(47) | 0.124 |
| 31–42 | 19.5(50) | | 18.6(38) | | 17.7(26) | |
| ≥43 | 23.8(61) | | 24.5(50) | | 19.7(29) | |
| **Sex** | | | | | | |
| Female | 84.0(215) | 0.720 | 89.2(182) | **<0.001** | 86.4(127) | **0.042** |
| Male | 16.0(41) | | 10.8(22) | | 13.6(20) | |
| **Marital status** | | | | | | |
| Married | 79.3(203) | 0.761 | 78.4(160) | 0.724 | 81.0(119) | 0.402 |
| Unmarried | 20.7(53) | | 21.6(44) | | 19.0(28) | |
| **Religion** | | | | | | |
| Muslim | 74.2(190) | | 72.1(147) | | 76.9(113) | |
| Hindu | 23.0(59) | 0.809 | 25.0(51) | 0.487 | 20.4(30) | 0.925 |
| Christian | 2.7(7) | | 2.9(6) | | 2.7(4) | |
| **Education** | | | | | | |
| Diploma nursing | 70.7(181) | | 72.5(148) | | 68.7(101) | |
| BSc nursing | 21.1(54) | **0.001** | 18.1(37) | **0.001** | 21.8(32) | 0.535 |
| Masters/above | 8.2(21) | | 9.3(19) | | 9.5(14) | |
| **Monthly income (Taka)** | | | | | | |
| ≤22400 | 10.9(28) | | 7.4(15) | | 8.2(12) | |
| 22401–33000 | 56.6(145) | **0.025** | 57.8(118) | **<0.001** | 59.2(87) | **0.014** |
| 33001–44000 | 10.9(28) | | 10.3(21) | | 10.9(16) | |
| 44001–55000 | 14.1(36) | | 15.2(31) | | 13.6(20) | |
| ≥55001 | 7.4(19) | | 9.3(19) | | 8.2(12) | |
| **Monthly family income (Taka)** | | | | | | |
| ≤22400 | 9.8(25) | | 9.3(19) | | 10.2(15) | |
| 22401–33000 | 9.4(24) | | 10.8(22) | | 10.2(15) | |
| 33001–44000 | 9.0(23) | 0.335 | 9.8(20) | 0.182 | 9.5(14) | 0.159 |
| 44001–55000 | 25.0(64) | | 24.0(49) | | 29.9(44) | |
| ≥55001 | 46.9(120) | | 46.1(94) | | 40.1(59) | |
| **Present designation** | | | | | | |
| Registered nurse | 7.4(19) | 0.701 | 8.3(17) | 0.527 | 10.2(15) | 0.135 |
| Nursing officer | 92.6(237) | | 91.7(187) | | 89.8(132) | |
| **Years of experience** | | | | | | |
| ≤2 | 46.9(120) | | 47.1(96) | | 49.7(73) | |
| 3–10 | 16.4(42) | | 15.2(31) | | 18.4(27) | |
| 11–18 | 10.9(28) | 0.506 | 11.3(23) | **0.036** | 10.2(15) | 0.249 |
| 19–26 | 14.1(36) | | 12.3(25) | | 12.2(18) | |
| ≥27 | 11.7(30) | | 14.2(29) | | 9.5(14) | |

studies conducted among nurses in Bangladesh also identified low compliance with IPC guidelines due to the lack of facilities, resource constraints, and heavy workload [28, 29]. These findings revealed that nurses in our study did not have sound knowledge about all IPC guidelines. While being asked about universal precautions, only 63% of nurses responded that they knew

**Table 6. Multivariable analysis of factors associated with levels of average knowledge, favourable attitude and safe practice toward infection prevention and control.**

| Characteristics | Average knowledge | | Favorable attitude | | Safe practice | |
|---|---|---|---|---|---|---|
| | AOR (95% CI) | p-value | AOR (95% CI) | p-value | AOR (95% CI) | p-value |
| **Age in years** [a] | | | | | | |
| ≤26 | 1.0 | | 1.0 | | 1.0 | |
| 27–30 | 2.5(1.0–6.5) | 0.057 | 1.8(0.9–3.7) | 0.106 | 1.3(0.6–2.5) | 0.506 |
| 31–42 | 0.8(0.3–2.0) | 0.712 | 0.6(0.2–1.7) | 0.326 | 0.4(0.2–1.0) | **0.049** |
| ≥43 | 0.6(0.2–2) | 0.415 | 0.2(0.0–1.0) | 0.053 | 0.1(0.0–0.2) | **<0.001** |
| **Sex** [b] | | | | | | |
| Female | 0.9(0.4–2.2) | 0.857 | 2.7(1.4–5.3) | **0.002** | 1.8(1.0–3.5) | 0.069 |
| Male | 1.0 | | 1.0 | | 1.0 | |
| **Marital status** [c] | | | | | | |
| Unmarried | 1.0(0.4–2.3) | 0.981 | 1.2(0.6–2.4) | 0.529 | 0.5(0.2–1.0) | **0.039** |
| Married | 1.0 | | 1.0 | | 1.0 | |
| **Religion** [d] | | | | | | |
| Hindu | 1.0(0.5–2.3) | 0.920 | 1.4(0.7–2.7) | 0.308 | 0.8(0.5–1.5) | 0.579 |
| Christian | 0.8(0.1–4.2) | 0.776 | 1.9(0.4–9.2) | 0.401 | 1.0(0.2–4.8) | 0.975 |
| Muslim | 1.0 | | 1.0 | | 1.0 | |
| **Education** [e] | | | | | | |
| Diploma nursing | 1.0 | | 1.0 | | 1.0 | |
| BSc nursing | 0.3(0.1–0.5) | **<0.001** | 0.3(0.2–0.6) | **<0.001** | 0.7(0.4–1.3) | 0.272 |
| Masters/above | 0.2(0.1–0.7) | **0.010** | 0.4(0.2–1.1) | 0.090 | 1.2(0.5–3.3) | 0.694 |
| **Monthly income (Taka)** [f] | | | | | | |
| ≤22400 | 1.0 | | 1.0 | | 1.0 | |
| 22401–33000 | 0.3(0.1–0.6) | **0.002** | 0.2(0.1–0.4) | **<0.001** | 0.2(0.1–0.5) | **<0.001** |
| 33001–44000 | 0.8(0.2–3.1) | 0.763 | 1.1(0.4–3.0) | 0.885 | 2.6(0.9–7.6) | 0.076 |
| 44001–55000 | 0.7(0.2–2.9) | 0.604 | 2.1(0.6–6.9) | 0.233 | 3.7(1.1–12.3) | **0.030** |
| ≥55001 | 2.3(0.2–24.4) | 0.499 | 14(1.3–147) | 0.028 | 5.8(1.3–26.1) | **0.022** |
| **Monthly family income** [g] **(Taka)** [g] | | | | | | |
| ≤22400 | 1.0 | | 1.0 | | 1.0 | |
| 22401–33000 | 3.4(0.7–16) | 0.128 | 2.2(0.8–6.2) | 0.119 | 2.5(0.9–6.3) | 0.064 |
| 33001–44000 | 5.6(0.7–45.1) | 0.106 | 5.3(1.4–20.4) | **0.014** | 2.6(1.0–6.8) | **0.045** |
| 44001–55000 | 1.8(0.5–6.1) | 0.354 | 2.8(1.0–7.9) | 0.058 | 2.7(1.0–7.3) | **0.045** |
| ≥55001 | 1.6(0.7–3.8) | 0.236 | 1.6(0.8–3.1) | 0.153 | 2.7(1.4–5.1) | **0.003** |
| **Present designation** [h] | | | | | | |
| Nursing officer | 1.0 | | 1.0 | | 1.0 | |
| Registered nurse | 0.8(0.2–2.4) | 0.631 | 1.5(0.5–4.5) | 0.441 | 2.0(0.7–5.5) | 0.169 |
| **Years of experience** [i] | | | | | | |
| ≤2 | 1.0 | | 1.0 | | 1.0 | |
| 3–10 | 0.9(0.5–1.9) | 0.867 | 0.9(0.4–2.1) | 0.842 | 1.0(0.5–2.2) | 0.976 |
| 11–18 | 0.6(0.2–1.5) | 0.276 | 0.6(0.2–2) | 0.436 | 0.8(0.2–2.4) | 0.661 |
| 19–26 | 0.1(0.1–0.4) | **<0.001** | 0.2(0–0.8) | **0.024** | 0.2(0.1–0.9) | **0.034** |

(*Continued*)

**Table 6.** (Continued)

| Characteristics | Average knowledge | | Favorable attitude | | Safe practice | |
|---|---|---|---|---|---|---|
| | AOR (95% CI) | p-value | AOR (95% CI) | p-value | AOR (95% CI) | p-value |
| ≥27 | 1.0(0.2–4.3) | 0.979 | 1.1(0.2–6.6) | 0.938 | 1.7(0.3–10.1) | 0.571 |

UOR = unadjusted odds ratios

AOR = adjusted odds ratios

[a] Knowledgeable adjusted for education, monthly income. Favorable attitude adjusted for sex, education, monthly income, monthly family income, total year of experience. Safe practice adjusted for monthly income, monthly family income.

[b] Knowledgeable adjusted for age, education, monthly family income. Favorable attitude adjusted for education, monthly income, total years of experience. Safe practice adjusted for monthly income, monthly family income.

[c] Knowledgeable adjusted for age, education, monthly income. Favorable attitude adjusted for sex, education, monthly income, monthly family income, total years of experience. Safe practice adjusted for age, monthly income, monthly family income

[d] Knowledgeable adjusted for age, education, monthly income. Favorable attitude adjusted for sex, education, monthly income, monthly family income, total years of experience. Safe practice adjusted for monthly income, monthly family income.

[e] Knowledgeable adjusted for age, monthly income. Favorable attitude adjusted for sex, monthly family income, total years of experience. Safe practice adjusted for monthly income, monthly family income.

[f] Knowledgeable adjusted for age, education. Favorable attitude adjusted for age, sex, education, monthly family income, total years of experience. Safe practice adjusted for age, monthly family income, total years of experience.

[g] Knowledgeable adjusted for age, education, monthly income. Favorable attitude adjusted for sex, education, monthly income, total years of experience. Safe practice adjusted for monthly income.

[h] Knowledgeable adjusted for age, education, monthly income. Favorable attitude adjusted for sex, education, monthly income, monthly family income, total years of experience. Safe practice adjusted for monthly income, monthly family income.

[i] Knowledgeable adjusted for education, monthly income. Favourable attitude adjusted for age, sex, education, monthly income, monthly family income. Safe practice adjusted for age, monthly income, monthly family income.

about WHO's '5 moments of hand hygiene' [30]. All healthcare workers have to be well aware of these moments, as hand hygiene is undisputedly the single most important measure for preventing the transmission of HAIs [31, 32]. The hospital leadership should take prompt precautionary measures that ensure regular training and periodic refreshers to impart awareness and training about hand hygiene [33].

It is recommended that the use of glutaraldehyde for 10 hours at 20–35˚C for sterilizing surgical and ward instruments is a recommended approach [34]. In our study, only 29% of the nurses knew how to properly use this chemical. All-round dissemination of information about its proper use would be helpful for nurses to take extra precautions while disinfecting the ward environment, particularly during this COVID-19 pandemic situation.

One of the major sources of HAIs in hospital settings is the recapping of previous needles. Especially this neglected tendency is more common in low-resource settings [35]. This attitude leads to greater incidences of needle stick injuries which can cause transmission of different blood-borne nosocomial infections among both patients and healthcare workers, such as Hepatitis B, C and HIV infection [9, 10]. Proper knowledge regarding recapping needles after use and before disposal is needed for nurses in the hospital setting. In our study, we found that the majority of nurses (95.7%) incorrectly described the process for disposing of used needles. This is important to dispose of the needles properly, and all nurses should be trained to dispose of used needles appropriately to avoid needlestick injuries.

Two-thirds (68%) of nurses had a favourable attitude toward IPC. Almost all the nurses showed a positive attitude towards policies and procedures for infection control. Moreover, the maximum number of nurses (93%) believed in the effectiveness of soap/hand sanitizer and the use of a gown/apron while dealing with potential body fluid exposure. This positivity and

remarkable adherence towards IPC practices are very much appreciable and markedly required to control infection transmission, especially concerning the ongoing COVID menace [36].

A significant number of nurses (approximately 65%) showed a favourable attitude towards the use of puncture-proof containers for the disposal of medical waste. HAI can be reduced to a certain extent if puncture-proof containers are regularly used for the disposal of medical wastes to reduce the threat of biomedical residues [37]. Proper evidence-based policy implementation along with useful training programmes would alert the nurses about the proper procedures for waste disposal.

So far, we have found the overall knowledge and attitude level of the majority of nurses towards IPC to be at an average level and a favourable stage (Table 3). Similar results were found in a study conducted in Iran [38]. However, in terms of practice, the overall score was not satisfactory (the mean score was <14 on a scale of 24). The study identified that a good portion of nurses (44.1%) did not follow safe IPC practices. Further component-based analysis revealed that about 60% of the nurses were aware of and failed to comply with the updated policies and guidelines related to infection control in facility premises. Almost half of the nurses (49.7%) did not receive vaccinations for common pathogens. This is quite alarming as without being vaccinated, nurses would face a greater risk of infection from commonly transmitted clinical pathogens, which would pose a threat to their patients, especially those who are immune-compromised [39]. On the other hand, vaccinated nurses can act as barriers against infection transmission, which would expedite essential healthcare service deliveries [40–42]. Only half of the nurses (49.3%) used masks and eye protection while performing procedures that were considered invasive or posed a risk of exposure to bodily fluids.

We also found some interesting findings by performing bi- and multivariable analysis to discover any potential association of the outcome variables with the covariates. We observed that experienced nurses had less favourable attitudes towards IPC. Similarly, nurses who were unmarried, low earners, older or more experienced exhibited less safe IPC practices. Previous studies conducted also inferred that earnings always greatly impact work performance and a better earning, healthy environment, as well as promotion opportunities, have a positive effect on the employee's attitude and practice in any work sector [43]. The study used a comprehensive tool to assess nurses' KAP, and this tool can be used and replicated in other settings and institutions by doing the necessary adaption.

We also would like to take the opportunity of discussing some potential limitations of this study. Firstly, the findings cannot be generalised across all hospital facilities, as this was only conducted in a tertiary care facility. Further rigorous studies would be able to address this issue by enrolling a larger number of study hospitals across the country. Secondly, possible over or under-reporting in one or more components of KAP towards standard precaution and IPC practices might bias the results, although comprehensive analyses were conducted to minimize such biases. Thirdly, nurses from all specialities and departments could not be enrolled due to time and resource limitations and it's already proven that traditional norms followed in a department might greatly influence nurses' practice compliance towards IPC measures. Finally, there are also chances of having socially accepted answers from the participants even though confidentiality and anonymity were ensured.

## Conclusion

The majority of nurses were knowledgeable and had a favourable attitude towards IPC, however they were lacking in safe IPC practices. Aged and experienced nurses were found more reluctant towards IPC. Policymakers and hospital leadership should provide nurses with

evidence-based recommendations and the necessary training to effectively implement IPC practices.

## Acknowledgments

The authors would like to sincerely thank the study participants for their precious time and commitment to complete the survey. The authors also express their cordial gratitude to Dhaka Medical College Hospital authorities for providing the necessary permission and logistic supports to conduct the study.

## Author Contributions

**Conceptualization:** Md. Golam Dostogir Harun, Md Mahabub Ul Anwar, Kusum Datta, Md. Imdadul Haque, A. B. M. Alauddin Chowdhury, Sabrina Sharmin, Md Saiful Islam.

**Data curation:** Md. Golam Dostogir Harun, Shariful Amin Sumon, Kusum Datta, Md. Imdadul Haque.

**Formal analysis:** Md Mahabub Ul Anwar, Shariful Amin Sumon, Md Abdullah-Al-Kafi, Md. Imdadul Haque, Sabrina Sharmin, Md Saiful Islam.

**Investigation:** Md. Golam Dostogir Harun, Md Mahabub Ul Anwar, Shariful Amin Sumon, Md Abdullah-Al-Kafi, Md. Imdadul Haque, Sabrina Sharmin.

**Methodology:** Md. Golam Dostogir Harun, Md Mahabub Ul Anwar, Shariful Amin Sumon, Md Abdullah-Al-Kafi, Kusum Datta, A. B. M. Alauddin Chowdhury, Sabrina Sharmin, Md Saiful Islam.

**Resources:** Md. Golam Dostogir Harun.

**Software:** Md Abdullah-Al-Kafi.

**Supervision:** Md. Golam Dostogir Harun, Md Mahabub Ul Anwar, A. B. M. Alauddin Chowdhury, Md Saiful Islam.

**Validation:** Md. Golam Dostogir Harun, Md Abdullah-Al-Kafi.

**Visualization:** Md. Golam Dostogir Harun, Md Mahabub Ul Anwar.

**Writing – original draft:** Md. Golam Dostogir Harun, Sabrina Sharmin, Md Saiful Islam.

**Writing – review & editing:** Md. Golam Dostogir Harun, Md Mahabub Ul Anwar, Shariful Amin Sumon, Md Abdullah-Al-Kafi, Kusum Datta, Md. Imdadul Haque, A. B. M. Alauddin Chowdhury, Md Saiful Islam.

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
