## [Decision Letter · Decision Letter 0]

28 Sep 2022

PONE-D-22-12028Pre-COVID-19 knowledge, attitude and practice among nurses towards infection prevention and control in Bangladesh: A hospital-based cross-sectional surveyPLOS ONE

Dear Dr. Harun,

Thank you for submitting your manuscript to PLOS ONE. After careful consideration, we feel that it has merit but does not fully meet PLOS ONE’s publication criteria as it currently stands. Therefore, we invite you to submit a revised version of the manuscript that addresses the points raised during the review process.

This is an interesting and valuable study of the infection control knowledge, attitudes, and practices among nurses at a large hospital in Bangladesh. While there are strengths of the study that make it worth publishing, the reviewers have identified several weaknesses that will need to be addressed. Please make the following required changes before resubmitting the manuscript:

1. Referring to comment #1 from Reviewer #1, please include information on whether the study site had an infection prevention and control program at the time the study was conducted.

2. Referring to comment #2 from Reviewer #1, please provide additional citations for the development of the questionnaire.

3. Referring to comment #3 from Reviewer #1, please specify in the methods section how the questionnaire was evaluated and what changes were made after the evaluation process.

4. Referring to comment #6 from Reviewer #1, it seems that the questionnaire has been translated into English, but the translation includes several errors that can create confusion in interpreting the results of the study. For example, it is unclear what is being asked in the item “All staff working in the hospital word /carefully deal with the patient should be considered potentially infection control guideline”. For the item “What resources/ supplies needed to comply with infection prevention guidelines”, it is not clear whether this was a generic question about whether nurses felt that they know what supplies are needed, or whether nurses were asked to identify specific supplies. A third example is the item “Bathing after sending time in infection suspected areas in hospital is for avoiding safety from infection”. As there are several more examples of confusing questionnaire items, the authors will need to review the translation of the questionnaire and ensure that it accurately represents the questions that were asked of the participants. The manuscript as a whole will also benefit greatly by being reviewed for grammar and spelling errors.

5. Please address the comment from Reviewer #2 regarding your questionnaire item that states that “Stethoscope must be cleaned/sterilized before and after every patient examine”.

6. Please address the comment from Reviewer #2 regarding the statement “it is recommended that the use of glutaraldehyde for at least 10 minutes for sterilizing surgical and ward instruments”.

We look forward to receiving your revised manuscript.

Kind regards,

Cindy Prins

Academic Editor

PLOS ONE

2.Thank you for stating the following financial disclosure:

“No”

3.Thank you for stating the following in your Competing Interests section: 

“No”

4.In your Data Availability statement, you have not specified where the minimal data set underlying the results described in your manuscript can be found. PLOS defines a study's minimal data set as the underlying data used to reach the conclusions drawn in the manuscript and any additional data required to replicate the reported study findings in their entirety. All PLOS journals require that the minimal data set be made fully available. For more information about our data policy, please see http://journals.plos.org/plosone/s/data-availability.

5.We note that you have indicated that data from this study are available upon request. PLOS only allows data to be available upon request if there are legal or ethical restrictions on sharing data publicly. For more information on unacceptable data access restrictions, please see http://journals.plos.org/plosone/s/data-availability#loc-unacceptable-data-access-restrictions.

6.Your ethics statement should only appear in the Methods section of your manuscript. If your ethics statement is written in any section besides the Methods, please move it to the Methods section and delete it from any other section. Please ensure that your ethics statement is included in your manuscript, as the ethics statement entered into the online submission form will not be published alongside your manuscript.

Reviewers' comments:

Reviewer's Responses to Questions

**Comments to the Author**

1. Is the manuscript technically sound, and do the data support the conclusions?

Reviewer #1: Yes

Reviewer #2: No

2. Has the statistical analysis been performed appropriately and rigorously? 

Reviewer #1: Yes

Reviewer #2: N/A

3. Have the authors made all data underlying the findings in their manuscript fully available?

Reviewer #1: No

Reviewer #2: Yes

4. Is the manuscript presented in an intelligible fashion and written in standard English?

Reviewer #1: Yes

Reviewer #2: Yes

5. Review Comments to the Author

Reviewer #1: In this well written paper, Harun and co-workers have constructed a questionnaire to investigate nurses' knowledge, attitudes and practices regarding infection prevention and control. The theme is timely and important. The statistical methods are sound and the results are clearly presented and easy to understand.

I have only minor comments and suggestions:

1. Study design and setting: Please write whether or not the institution had impelemented an infection control and prevention programme at the time of the study. If ypu like, the descritpion of the progamme could e.g. be compared to the core competencies of infection prevention and control as described by WHO: https://apps.who.int/iris/bitstream/handle/10665/335821/9789240011656-eng.pdf. Maybe this is a relevant reference for justifying some of the questions you ended up wih.

2. Methods, line 133-135: The authors state that the questionnaire was developed based on litterature reviews and expert opinion. The only reference to litterature is reference 25, a master thesis. Please state examples of other litterature sources (e. g. I see that one of the questions refers to WHO's "5 moments for hand hygiene").

3. Line 135: How was the questionnaire evaluated? Were there made adjustments after evaluation? Please add this information to the manuscript.

4. Line 162: Educational level was also part of the analyses and should be added along side socioeconomic and demographic variables.

5. Discussion: it would be interesting to hear whether or not you would recommend this tool to be used in other instiutions, whether it could be used internationally and what adaptions that might be necessary for a broader use.

6. Appendix. Please check the translation of the questionnaire as there are several mis-spellings and incomplete sentences.

Please check spelling in line 39, 89, 230, 303 and table 2 (monthly family ...) and in the translated questionnaire.

Reviewer #2: Dear authors

We believe that this research question you have embraced is extremely important. It is crucial to explore and understand Knowledge, attitudes and practices related to the prevention of healthcare acquired infections as a means of improving educational approaches. In this regard, we encourgae you to continue the research.

However, this work has important flaws. There are mistaken concepts of healthcare infection prevention. As examples, in Table4, section "universal precautions" you state that stethoscopes should be cleaned/sterilized before and after patient "examine". Stethoscopes are considered non critical items as they get in touch with the skin of patients, so they need low level disinfection after the patient examination. If there are skin lesions or the patient is in contact precautions due to colonization/infection by a multidrug resistant pathogen, non critical items should be dedicated to the patient (preferably, if you have such items available). Another example of misconception is the exposition time of medical devices/equipments to glutaraldehyde. In line 297 you state that "it is recommended that the use of glutaraldehyde for at least 10 minutes for sterilizing surgical and ward instruments". This is not correct. For chemical sterilization it is required 8 to 10 hours of exposition of glutaraldehyde. I suggest a look at https://www.cdc.gov/infectioncontrol/guidelines/disinfection/tables/table1.html for the revision of methods of sterilization and disinfection of medical items.

Table 4 is intended to scrutinize Knowledge of nurses about HAIs prevention. However, they have generic questions about the existance of guidelines and do not explore the knowledge of principles of infection prevention like routes of pathogen transmission, universal and specific precautions (aerossol/droplets), modes of cleaning/disinfection/sterilization of items, immunization, good practices of invasive devices care (ex. central venous catheters), reuse of items (syringes, needles), waste management just to stay with some examples. Thechnical concepts of infection should be thoroughly revised.

6. PLOS authors have the option to publish the peer review history of their article (what does this mean?). If published, this will include your full peer review and any attached files.

Reviewer #1: No

Reviewer #2: No

---

## [Author Response · Author response to Decision Letter 0]

19 Oct 2022

Date: October 16, 2022

To 

The Editor

PLOS ONE

Subject: Incorporation of reviewers’ comments and resubmission of the manuscript (ID PONE-D-22-12028) titled "Pre-COVID-19 knowledge, attitude and practice among nurses towards infection prevention and control in Bangladesh: A hospital-based cross-sectional survey"

Dear Editor:

Thank you very much for your email regarding the incorporation of reviewers' comments on our manuscript (PONE-D-22-12028) titled "Pre-COVID-19 knowledge, attitude and practice among nurses towards infection prevention and control in Bangladesh: A hospital-based cross-sectional survey."

I am pleased to resubmit the revised version. We (Author & Co. Authors) tried our best to concentrate on comments/suggestions made by both reviewers. We appreciated both of the reviewers for their excellent comments. All comments/suggestions have been incorporated into the manuscript as well. 

We hope that the incorporated comments/suggestion will satisfy the reviewers and editor and expect the continuation of the production of our manuscript.

Editor comments and authors' response:

This is an interesting and valuable study of the infection control knowledge, attitudes, and practices among nurses at a large hospital in Bangladesh. While there are strengths of the study that make it worth publishing, the reviewers have identified several weaknesses that will need to be addressed. Please make the following required changes before resubmitting the manuscript:

Authors’ response: 

Thank you so much for your constructive comment and guidance. We have addressed the point-by-point comments raised by both reviewers and revised the manuscript accordingly. 

Editor comments-1: Referring to comment #1 from Reviewer #1, please include information on whether the study site had an infection prevention and control program at the time the study was conducted.

Authors’ Response: 

Thank you. No active infection prevention and control program was running in our study site during the study period to assess the current infection level and take mitigation strategies. Also, thank you so much for sharing the WHO ‘CORE COMPETENCIES FOR INFECTION PREVENTION AND CONTROL PROFESSIONALS’ document. We have rechecked and found no active IPC program was running during our study period.

Editor comments-2: Referring to comment #2 from Reviewer #1, please provide additional citations for the development of the questionnaire.

Authors’ Response: 

Thank you so much for the valuable suggestions. We have included two more citations, including ‘WHO 5 moments, in the revised manuscript. 

Editor comments-3: Referring to comment #3 from Reviewer #1, please specify in the methods section how the questionnaire was evaluated and what changes were made after the evaluation process.

Authors’ Response: 

This questionnaire was pretested among the 20 non-sampling nurses (who were excluded from the sampling frame) using the Bangla version of the questionnaire to get feedback on the suitability, appropriateness, and sequencing of the questions. We addressed the feedback, made the language easier, updated the questionnaire, and conducted the KAP survey among nurses. We have incorporated this in the method section and revised the manuscript.

Editor comments-4: Referring to comment #6 from Reviewer #1, it seems that the questionnaire has been translated into English, but the translation includes several errors that can create confusion in interpreting the results of the study. For example, it is unclear what is being asked in the item “All staff working in the hospital word /carefully deal with the patient should be considered potentially infection control guideline”. For the item “What resources/ supplies needed to comply with infection prevention guidelines”, it is not clear whether this was a generic question about whether nurses felt that they know what supplies are needed, or whether nurses were asked to identify specific supplies. A third example is the item “Bathing after sending time in infection suspected areas in hospital is for avoiding safety from infection”. As there are several more examples of confusing questionnaire items, the authors will need to review the translation of the questionnaire and ensure that it accurately represents the questions that were asked of the participants. The manuscript as a whole will also benefit greatly by being reviewed for grammar and spelling errors.

Authors’ Response: 

Thank you so much for your kind and very important observation. We are very sorry that there were some spelling and grammatical mistakes in the annex questions. We carefully reviewed the questionnaire and corrected the document in the revised manuscript

Editor comments-5: Please address the comment from Reviewer #2 regarding your questionnaire item that states that “Stethoscope must be cleaned/sterilized before and after every patient examine”

Authors’ Response: 

Thank you so much for the observation: We fully agree with the reviewer that Stethoscopes are considered non-critical items as they get in touch with the skin of patients, so they need low-level disinfection after the patient examination. We have rechecked the question and revised it to “The stethoscope must be cleaned/sanitized with an alcohol swab pad before and after every patient examines” and incorporated it in the revised manuscript. 

Editor comments-6: Please address the comment from Reviewer #2 regarding the statement “it is recommended that the use of glutaraldehyde for at least 10 minutes for sterilizing surgical and ward instruments”.

Authors’ Response

This is a great observation, and we are sorry for the mistake. Also, thank you for sharing the CDC guidelines link. We have reviewed the documents and corrected the statement with new references in the revised manuscript. 

Reviewers' comments and Authors' responses

Authors’ response: We sincerely thank the reviewers for their attention and the constructive comments they have made to strengthen the paper. Please see our reflections below, preceded by “Response .”We hope our responses and edits have strengthened the paper and look forward to your assessment of this revision.

Reviewer # 1 comments and authors’ response: 

General cpmments: 

In this well written paper, Harun and co-workers have constructed a questionnaire to investigate nurses' knowledge, attitudes and practices regarding infection prevention and control. The theme is timely and important. The statistical methods are sound and the results are clearly presented and easy to understand. I have only minor comments and suggestions:

Authors’ Response: 

Thank you so much to the reviewer for your appreciation and positive remarks. We also appreciate your kind effort and time in providing very important comments. Your comments and feedback will enrich our manuscript significantly. We are delighted to address your remarks accordingly

Comments: 1. Study design and setting: Please write whether or not the institution had impelemented an infection control and prevention programme at the time of the study. If ypu like, the descritpion of the progamme could e.g. be compared to the core competencies of infection prevention and control as described by WHO: https://apps.who.int/iris/bitstream/handle/10665/335821/9789240011656-eng.pdf. Maybe this is a relevant reference for justifying some of the questions you ended up wih.

Authors’ Response: 

Thank you. No active infection prevention and control program was running in our study site during the study period to assess the current infection level and take mitigation strategies. Also, thank you so much for sharing the WHO ‘CORE COMPETENCIES FOR INFECTION PREVENTION AND CONTROL PROFESSIONALS’ document. We have rechecked and found no active IPC program was running during our study period.

2. Methods, line 133-135: The authors state that the questionnaire was developed based on litterature reviews and expert opinion. The only reference to litterature is reference 25, a master thesis. Please state examples of other litterature sources (e. g. I see that one of the questions refers to WHO's "5 moments for hand hygiene").

Authors’ Response: 

Thank you so much for the suggestions. We have included two more citation including ‘WHO 5 moments’ in the revised manuscript. 

3. Line 135: How was the questionnaire evaluated? Were there made adjustments after evaluation? Please add this information to the manuscript.

Authors’ Response: 

This questionnaire was pretested among the 20 non-sampling nurses (who were excluded from the sampling frame) using the Bangla version of the questionnaire to get feedback on the suitability, appropriateness, and sequencing of the questions. We addressed the feedback, made the language more easy, updated the questionnaire, and conducted the KAP survey among nurses.

4. Line 162: Educational level was also part of the analyses and should be added along side socioeconomic and demographic variables.

Authors’ Response: 

Thank you so much for the suggestions. We have incorporated the education level in the socioeconomic and demographic section and revised the manuscript accordingly.

5. Discussion: it would be interesting to hear whether or not you would recommend this tool to be used in other instiutions, whether it could be used internationally and what adaptions that might be necessary for a broader use.

Authors’ Response: 

Thank yous o much for the suggestion. We have added a sentence to recomnd that the study used a comprehensive tool to assess nurses' KAP, and this tool can be used and replicated in other settings and institutions by doing the necessary adaption. 

6. Appendix. Please check the translation of the questionnaire as there are several mis-spellings and incomplete sen

Authors’ Response: 

Thank you so much for your kind and very important observation. We are very sorry that there were some spelling and grammatical mistakes in the annex questions. We carefully reviewed the questionnaire and corrected the document in the revised manuscript

7. Please check spelling in line 39, 89, 230, 303 and table 2 (monthly family ...) and in the translated questionnaire.

Authors’ Response: 

Thank you so much for your kind observation. We have checked and corrected the spelling.

Reviewer # 2 comments and authors’ response: 

Reviewer #2: Dear authors

We believe that this research question you have embraced is extremely important. It is crucial to explore and understand Knowledge, attitudes and practices related to the prevention of healthcare acquired infections as a means of improving educational approaches. In this regard, we encourgae you to continue the research.

However, this work has important flaws. There are mistaken concepts of healthcare infection prevention. 

As examples, in Table4, section "universal precautions" you state that stethoscopes should be cleaned/sterilized before and after patient "examine". Stethoscopes are considered non critical items as they get in touch with the skin of patients, so they need low level disinfection after the patient examination. If there are skin lesions or the patient is in contact precautions due to colonization/infection by a multidrug resistant pathogen, non critical items should be dedicated to the patient (preferably, if you have such items available). 

Authors’ Response: 

Thank you so much for the observation: We fully agree with the reviewer that Stethoscopes are considered non-critical items as they get in touch with the skin of patients, so they need low-level disinfection after the patient examination. We have rechecked the question and revised it as “The stethoscope must be cleaned/sanitized with an alcohol swab pad before and after every patient examines” and incorporated it in the revised manuscript. 

Reviewer comments: Another example of misconception is the exposition time of medical devices/equipments to glutaraldehyde. In line 297 you state that "it is recommended that the use of glutaraldehyde for at least 10 minutes for sterilizing surgical and ward instruments". This is not correct. For chemical sterilization it is required 8 to 10 hours of exposition of glutaraldehyde. I suggest a look at https://www.cdc.gov/infectioncontrol/guidelines/disinfection/tables/table1.html for the revision of methods of sterilization and disinfection of medical items.

Authors’ Response: 

This is a great observation, and we are sorry for the mistake. Also, thank you for sharing the CDC guidelines link. We have reviewed the documents and corrected the statement with new references in the revised manuscript. 

Table 4 is intended to scrutinize Knowledge of nurses about HAIs prevention. However, they have generic questions about the existance of guidelines and do not explore the knowledge of principles of infection prevention like routes of pathogen transmission, universal and specific precautions (aerossol/droplets), modes of cleaning/disinfection/sterilization of items, immunization, good practices of invasive devices care (ex. central venous catheters), reuse of items (syringes, needles), waste management just to stay with some examples. Thechnical concepts of infection should be thoroughly revised.

Authors’ Response: 

These are excellent suggestions. We have incorporated the reviewer's advice and revised the questionnaire and manuscript accordingly. We did not include the west management questions in the KAP as nurses are not involved in the hospital's west management activities. A separate team in Bangladeshi hospitals is responsible for the west management activities, and the different groups also supervise them. So we excluded questions related to west management.

---

## [Editor Report · Decision Letter 1]

28 Oct 2022

PONE-D-22-12028R1Pre-COVID-19 knowledge, attitude and practice among nurses towards infection prevention and control in Bangladesh: A hospital-based cross-sectional surveyPLOS ONE

Dear Dr. Harun,

Thank you for submitting your manuscript to PLOS ONE. After careful consideration, we feel that it has merit but does not fully meet PLOS ONE’s publication criteria as it currently stands. Therefore, we invite you to submit a revised version of the manuscript that addresses the points raised during the review process.

Thank you for revising and resubmitting your manuscript, “Pre-COVID-19 knowledge, attitude and practice among nurses towards infection prevention and control in Bangladesh: A hospital-based cross-sectional survey”. Most of the reviewers' comments have been addressed adequately with the exception of two issues.

The first issue relates to the knowledge item about glutaraldehyde sterilization. In the original manuscript, responses to the survey item “A contaminated item soaked in glutaraldehyde for 10 minutes is sterilized” were listed as 29% “Correct” and 71% “Incorrect”. Reviewer #2 had commented that “In line 297 you state that "it is recommended that the use of glutaraldehyde for at least 10 minutes for sterilizing surgical and ward instruments". This is not correct. For chemical sterilization it is required 8 to 10 hours of exposition of glutaraldehyde.”

The text in the original manuscript’s results section (lines 202 – 206) stated, “On the contrary, a low percentage of correct answers were found for knowledge questions related to the resources needed to comply with infection prevention guidelines (23.3%), bending used needles (4.3%), **drenching in glutaraldehyde for 10 minutes for decontamination (29.0%)** and compliance with IPC guideline even in heavy workload (40.3%)”.

This indicates that the responses to this survey item were originally considered to be correct if a respondent agreed that an item could be sterilized if soaked in glutaraldehyde for 10 minutes. In the revised manuscript, the authors added the following to the discussion (lines 298-300), “It is recommended that the use of glutaraldehyde for 10 hours at 20-35°C for sterilizing surgical and ward instruments is a recommended approach (34). In our study, only 29% of the nurses knew how to properly use this chemical.” However, the survey item still contains the original statement “A contaminated item soaked in glutaraldehyde for 10 minutes is sterilized” and the response rates are the same as they were in the previous version. There is a major concern about what time frame the survey item originally stated (10 minutes versus 10 hours) and why the percentage of “correct” and “incorrect” responses remains the same in the revised manuscript. If a respondent answered that 10 minutes was not adequate then they actually had the right answer, which would indicate that 71% of respondents were “Correct” instead of “Incorrect”. The authors must address this issue and assure that those results are presented consistently and accurately throughout the manuscript.  

The second issue that needs to be addressed is the survey item “All staff working in the hospital should follow the IPC instructions and carefully deal with the patient is considered a potential infection control practice”. The translation of this item to English is confusing and thus the meaning of the item is unclear.  The first part of the item (All staff working in the hospital should follow the IPC instructions) does not need to be changed but the second part of the item (and carefully deal with the patient is considered a potential infection control practice) needs to be edited for clarity.

We look forward to receiving your revised manuscript.

Kind regards,

Cindy Prins

Academic Editor

PLOS ONE
---

## [Author Response · Author response to Decision Letter 1]

10 Nov 2022

Date: October 30, 2022

To 

The Editor in Chief 

PLOS ONE

Subject: Incorporation of reviewers’ comments and resubmission of the manuscript (ID PONE-D-22-12028) titled "Pre-COVID-19 knowledge, attitude and practice among nurses towards infection prevention and control in Bangladesh: A hospital-based cross-sectional survey"

Dear Editor:

Thank you very much for your email regarding the incorporation of reviewers' comments on our manuscript (PONE-D-22-12028) titled "Pre-COVID-19 knowledge, attitude and practice among nurses towards infection prevention and control in Bangladesh: A hospital-based cross-sectional survey."

I am pleased to resubmit the revised version. We (Author & Co. Authors) tried our best to concentrate on comments/suggestions made by both reviewers. We appreciated both of the reviewers for their excellent comments. All comments/suggestions have been incorporated into the manuscript as well. 

We hope that the incorporated comments/suggestion will satisfy the reviewers and editor and expect the continuation of the production of our manuscript.

Editor comments and authors' response:

Editor comments:The first issue relates to the knowledge item about glutaraldehyde sterilization. In the original manuscript, responses to the survey item “A contaminated item soaked in glutaraldehyde for 10 minutes is sterilized” were listed as 29% “Correct” and 71% “Incorrect”. Reviewer #2 had commented that “In line 297 you state that "it is recommended that the use of glutaraldehyde for at least 10 minutes for sterilizing surgical and ward instruments". This is not correct. For chemical sterilization it is required 8 to 10 hours of exposition of glutaraldehyde.”

The text in the original manuscript’s results section (lines 202 – 206) stated, “On the contrary, a low percentage of correct answers were found for knowledge questions related to the resources needed to comply with infection prevention guidelines (23.3%), bending used needles (4.3%), drenching in glutaraldehyde for 10 minutes for decontamination (29.0%) and compliance with IPC guideline even in heavy workload (40.3%)”.

This indicates that the responses to this survey item were originally considered to be correct if a respondent agreed that an item could be sterilized if soaked in glutaraldehyde for 10 minutes. In the revised manuscript, the authors added the following to the discussion (lines 298-300), “It is recommended that the use of glutaraldehyde for 10 hours at 20-35°C for sterilizing surgical and ward instruments is a recommended approach (34). In our study, only 29% of the nurses knew how to properly use this chemical.” However, the survey item still contains the original statement “A contaminated item soaked in glutaraldehyde for 10 minutes is sterilized” and the response rates are the same as they were in the previous version. There is a major concern about what time frame the survey item originally stated (10 minutes versus 10 hours) and why the percentage of “correct” and “incorrect” responses remains the same in the revised manuscript. If a respondent answered that 10 minutes was not adequate then they actually had the right answer, which would indicate that 71% of respondents were “Correct” instead of “Incorrect”. The authors must address this issue and assure that those results are presented consistently and accurately throughout the manuscript. 

Authors’ response: 

Thank you so much for your kind observation. Actually, we used the question “A contaminated item soaked in glutaraldehyde for 10 minutes is sterilized,” and those who answered ‘no’ was correct, and 29% answered correctly. However, we have presented this finding in the manuscript correctly. “A contaminated item soaked in glutaraldehyde for 10 hours at 20-35°C is sterilized”. We have also updated the result, discussion, and table sections accordingly. 

Editor comments: The second issue that needs to be addressed is the survey item “All staff working in the hospital should follow the IPC instructions and carefully deal with the patient is considered a potential infection control practice”. The translation of this item to English is confusing and thus the meaning of the item is unclear. The first part of the item (All staff working in the hospital should follow the IPC instructions) does not need to be changed but the second part of the item (and carefully deal with the patient is considered a potential infection control practice) needs to be edited for clarity.

Authors’ Response: 

Thank you for the observation. We have edited the question and made it clear for understanding in the revised version of the manuscript.

---

## [Editor Report · Decision Letter 2]

16 Nov 2022

Pre-COVID-19 knowledge, attitude and practice among nurses towards infection prevention and control in Bangladesh: A hospital-based cross-sectional survey

PONE-D-22-12028R2

Dear Dr. Harun,

We’re pleased to inform you that your manuscript has been judged scientifically suitable for publication and will be formally accepted for publication once it meets all outstanding technical requirements.

Kind regards,

Cindy Prins

Academic Editor

PLOS ONE
---

## [Editor Report · Acceptance letter]

21 Nov 2022

PONE-D-22-12028R2 

Pre-COVID-19 knowledge, attitude and practice among nurses towards infection prevention and control in Bangladesh: A hospital-based cross-sectional survey. 

Dear Dr. Harun:

I'm pleased to inform you that your manuscript has been deemed suitable for publication in PLOS ONE. Congratulations! Your manuscript is now with our production department. 

Kind regards, 

on behalf of

Dr. Cindy Prins 

Academic Editor

PLOS ONE